# Assessing Alternative Supporting Organic Materials for the Enhancement of Water Reuse in Subsurface Constructed Wetlands Receiving Acid Mine Drainage

Martha M. Oberholzer [1],*, Paul J. Oberholster [1], Luyanda L. Ndlela [2], Anna-Maria Botha [3] and Johannes C. Truter [4]

1   Centre for Environmental Management, University of the Free State, Bloemfontein 9300, South Africa; oberholsterpj@ufs.ac.za
2   Council for Scientific and Industrial Research, Pretoria 0001, South Africa; lndlela@csir.co.za
3   Department of Genetics, University of Stellenbosch, Stellenbosch 7600, South Africa; ambo@sun.ac.za
4   Department of Microbiology, University of Stellenbosch, Stellenbosch 7600, South Africa; jctruter@sun.ac.za
*   Correspondence: thia.viljoen@gmail.com; Tel.: +27-82-610-9481

**Abstract:** Acid mine drainage (AMD) is a global problem with severe consequences for the environment. South Africa's abandoned mines are a legacy from the country's economic dependence on the mining sector, with consequent negative impacts on ecosystems. AMD remediation includes active and passive techniques. Constructed wetlands (a passive technique) have lower operational costs but require larger spaces and longer timeframes to achieve the remediation of AMD, and are supported by anaerobic sulphate-reducing bacteria (SRB), which capable of remediating high-sulphate-laden AMD while precipitating dissolved metals from the AMD. Organic substrates supporting these activities are often the limiting factor. When enhancing existing passive AMD remediation technologies, alternative waste material research that may support SRB activity is required to support the circular economy through the reduction in waste products. Chicken feathers show potential as a substrate enhancer, boosting organic carbon availability to SRB, which sustains passive AMD treatment processes by achieving pH elevation, sulphate and metal reductions in AMD water for reuse. Microbial biodiversity is essential to ensure the longevity of passive treatment systems, and chicken feathers are proven to have an association with SRB microbial taxa. However, the longer-term associations between the AMD water parameters, microbial diversity and the selected substrates remain to be further investigated.

**Keywords:** acid mine drainage; sulphate-reducing bacteria; passive remediation

## 1. Introduction

Acid Mine Drainage (AMD) causes damage to global ecosystems with serious consequences for the receiving environment. In South Africa, AMD has been responsible for severe aquatic ecosystem damage and unusable land. Associated impacts include elevated cancer risks and deaths resulting from exposure to radioactive compounds and the dissolved hazardous metals associated with AMD. The lowered pH levels and increased sulphate ($SO_4$), salt and metal content of AMD contaminate surface water and groundwater resources which, in turn, have an impact on aquatic life and soil properties [1–9].

Sulphide minerals, such as pyrite, are typically present in coal mining areas and are exposed to the atmosphere as a result of mining practices and excavations. AMD is an effluent that results from the oxidation of sulphide-bearing rock, subsequently leaching into surface water. AMD sources are, however, not all linked to primary mining activities, but also originate from secondary sources including stockpiles, concentrated spillages, emergency ponds and treatment sludge ponds [10]. Ongoing mining activity, decant

from abandoned mines, economic significance and poor management practices present a challenge when curbing the production and prevention of AMD.

There are numerous technologies that can improve AMD water quality. These can be broadly divided into active and passive treatment strategies, which are further divided into biotic and abiotic interventions [7,10,11]. Active remediation involves the addition of alkaline materials, elevating the pH of the AMD and reducing the metal content through absorption, precipitation or coprecipitation. Generally, active treatments are linked to high capital costs for power resources and intensive labour requirements for its long-term maintenance [2,5,11–13]. Passive treatment systems are dependent on biological, geochemical, and physical processes. Different water pollutants can also be removed by passive permeable reactive barriers, specifically for the passive treatment of contaminated groundwater [14,15]. AMD treatment in passive systems is primarily achieved by sulphate-reducing bacteria (SRB) in controlled environments, which include wetlands, bioreactors, or geochemical systems capable of supporting biotic and abiotic reactions within AMD water. Microbial remediation is achieved by the production of alkalinity through the anaerobic biological reactions that occur during the consumption of organic carbon substrates and buffering free-hydrogen ions in the water. Sulphate ions are used by SRB as a catalyst in this reaction and are reduced to hydrogen sulphide, while the pH elevations result in metal precipitation [5,13]. A continuous supply of organic carbon substrates is essential to support the ongoing activities of SRB in passive treatment systems [16].

The passive treatment of AMD can be achieved using natural or constructed wetlands that provide numerous benefits, as highlighted by Pat-Espadas et al. [17] and Rambuda et al. [7]. Improvements in water quality are achieved through a variety of processes, including plant-mediated and microbial activities, chemical and physical processes [12]. These treatment systems are lower in cost, do not require electrical power sources and generally have lowered operational costs; however, extended timeframes are required to achieve water quality improvements, and larger operational areas are inevitable [12,18]. A variety of readily available organic substrate material, as well as waste substrates, have been used for AMD treatment in constructed wetlands. The selected substrates provide attachment sites for microorganisms that perform essential functions, as well as metals and nutrient sorption sites [19]. In constructed wetlands, waste substrates, including cow, chicken and sheep manure, wood chips, sawdust, peat, hay, sewage sludge, methanol, ethanol, compost, fly ash, slag, zerovalent iron particles and kiln waste, have been successfully and efficiently applied for AMD treatment [10,19]. However, in a quest to enhance existing AMD passive remediation technologies, which can form part of the circular economy, it is necessary to explore alternative, less explored, possibly longer-lasting, inexpensive and easily obtainable organic carbon sources that can sustain the various anaerobic microorganisms responsible for reducing the $SO_4$ concentrations in AMD. Determining the suitability of less explored, readily available alternative waste substrates or mixed substrates in the passive treatment technologies may lead to the more cost-effective implementation of passive AMD treatment facilities. The knowledge gained could enhance waste beneficiation and mitigate the environmental impacts associated with AMD. The objectives of the current study were as follows: (1) To evaluate the carbon content of selected organic materials to understand their suitability to support SRB growth. (2) To evaluate the impact of the selected substrates on the remediated AMD water quality for reuse. (3) To evaluate the microbial biodiversity supported by the various substrates after exposure to AMD water.

## 2. Materials and Methods

### 2.1. Sample Collections

#### 2.1.1. Acid Mine Drainage

AMD water was collected from the eMalahleni water reclamation plant (EWRP) situated near the city of Witbank (eMalahleni) in the province of Mpumalanga (Lat: 25°56′30.13″ S; Long: 29°11′38.69″ E), South Africa. This reclamation plant was commissioned in 2007 and actively treats AMD to potable water in large volumes [20]. Although

the AMD water received at the EWRP arrives from different coal collieries in the region, its general water quality parameters mimic typical AMD water (Figure 1). The AMD water used in the experimental set-up for both phases was obtained from this active treatment plant at the sample point dedicated to the retrieval of laboratory test samples for incoming AMD (pre-treatment). Cumulatively, 100 L of AMD water samples were collected in 1-L clean plastic water sample bottles for each phase and transported in cooler boxes with ice packs during commutation. The samples were subsequently stored at 4 °C in a fridge prior to use.

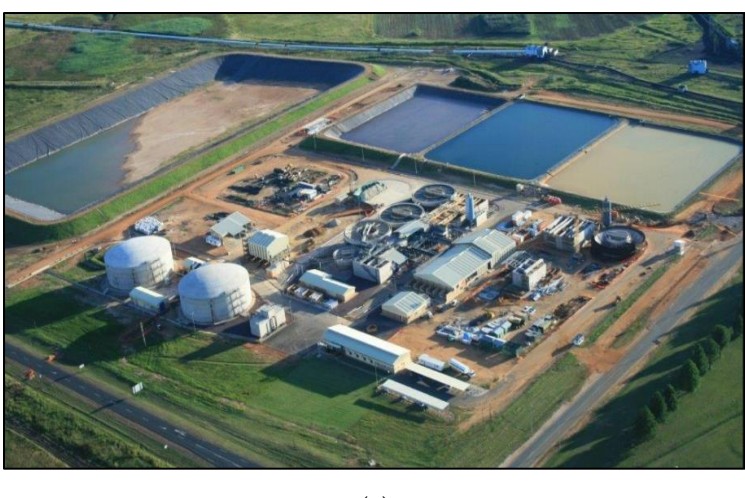

| Parameter | Concentration |
|---|---|
| pH | 3 |
| EC (mS/m) | 463 |
| TDS (mg/L) | 4442 |
| Ca (mg/L) | 517 |
| Mg (mg/L) | 245 |
| Na (mg/L) | 31 |
| K (mg/L) | 9 |
| SO$_4$ (mg/L) | 3266 |
| Cl (mg/L) | 10 |
| Fe (mg/L) | 255 |
| Mn (mg/L) | 25 |
| Al (mg/L) | 70 |

(**a**)                                              (**b**)

**Figure 1.** AMD active treatment plant at eMalahleni and its typical water quality parameters of untreated AMD; (**a**) image source from Infrastructure News [21]; (**b**) data from chemical analyses of AMD sampled.

2.1.2. Substrates

Readily available waste substrates were investigated as possible substrates for AMD remediation. These included cow manure, slimes from steelmaking processes (hereafter slimes), fly ash, mill scale sludge (hereafter sludge), metallurgical slag (hereafter slag), coal discard (hereafter discard) and chicken feathers. The cow manure was sourced from a local cattle pen in the Witbank area and was selected to serve as a control, considering the existing information related to the AMD treatment available for this material [19]. The slimes material originated from a steelmaking industry near the city of Witbank. The off-gas particulates of the abatement equipment resulting from the industrial processes were slimed by adding water and discarded onto slimes dams. Fly ash was obtained from a colliery near the town of Sasolburg and considered a waste material from this colliery's operations. The sludge originated from steel rolling mill process at a steelworks near the city of Witbank and was considered as waste originating from the rolling operations. Slag material was obtained from a steelmaking operation near the city of Witbank, originating from the ironmaking process. Ultrafine material originating from coal wash plants, discarded at coal tailings facilities, were sourced in the Witbank area. The chicken feathers were considered nuisance waste from the poultry abattoirs and were sourced from small-scale chicken vendors in the Western Cape province, South Africa.

*2.2. AMD Column eExposure*

2.2.1. Sample Preparation

Untreated AMD sourced from the EWRP was exposed to the various substrates selected (cow manure, slimes, sludge, fly ash, slag, discard, and chicken feathers) over a 35-day (Phase 1) and 73-day (Phase 2) timeframe. Individual and mixed substrates were applied, respectively, in Phase 1 and 2. The various substrates required certain sample preparation steps prior to their exposure to AMD in both phases. All samples were dried

in a drying oven at 65 °C for 48 h. The cow manure was then hand-milled with a porcelain mortar and pestle. The slimes, fly ash, sludge, slag and discard were screened and crushed (if needed) to a size of ~2 mm. The chicken feathers were rinsed with tap water and then with de-ionised water in batches to remove any debris (including sand, stone, glass and plant material), and dried in a drying oven at 65 °C for 48 h. The chicken feathers were then milled for 30 s in an industrial-scale sample preparation milling pot to obtain a fine chicken feather powder, which was suitable for use in the experimental columns (Figure 2).

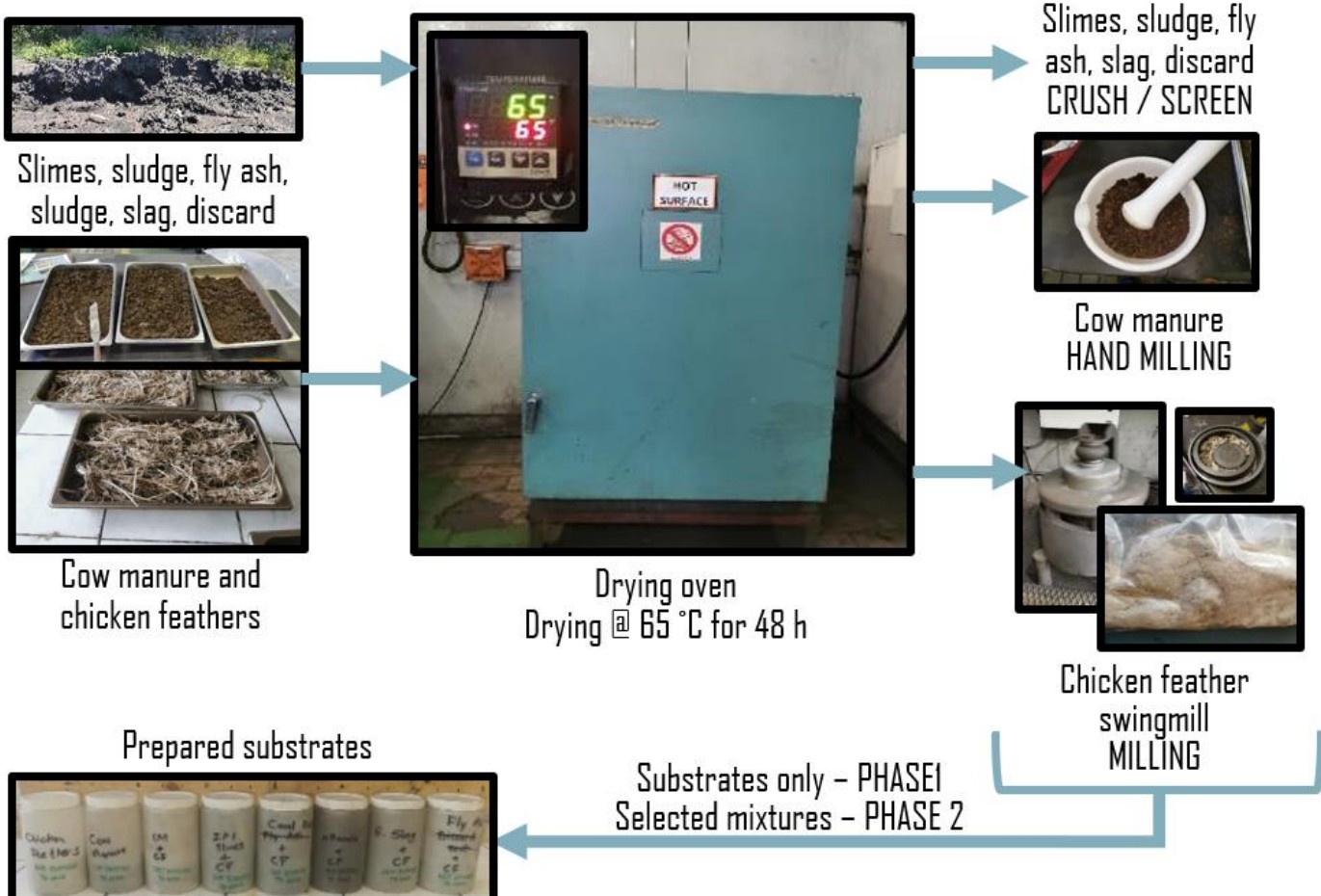

**Figure 2.** Substrate preparation steps taken prior to untreated AMD exposure.

2.2.2. Experimental Columns

The experiment was set up in two phases using lab-scale experimental columns (30-cm length with 10-cm diameter) in which an untreated AMD water sample ($\pm$2 L volume during Phase 1 and $\pm$2.5 L volume during Phase 2) was exposed to the selected organic materials in triplicate. Figure 3 demonstrates the typical composition of the experimental columns throughout the two experimental phases. Each column was then cleaned and rinsed three times with de-ionised water. The bottom part of each column was packed with coarse geotextile filter fabric (bottom), followed by lab-grade cotton wool (middle) and lab-grade glass wool (top) (sourced from LECO Africa). This prevented the clogging of the bottom drainage tap from which the treated/remediated AMD would be extracted at the end of exposure. The experiment was set up as a static exposure system, with no water flow through the system. The untreated AMD water was added to the packed substrate in the experimental column and left to stand static for the selected period for each phase. The experimental columns differed in layout through differences in exposure time, substrate contents, substrate volumes and AMD volumes in Phase 1 and 2. Table 1 summarises these differences. Both Phase 1 and 2 were executed at room temperature ($\pm$24 °C). The volumes

of substrates and substrate mixtures were kept constant throughout the different phases, although it must be noted that the substrate densities significantly differed based on the nature of the substrates and substrate mixtures.

During Phase 1, untreated AMD was exposed to the various selected substrates. The prepared substrates were taken as is, weighed, and measured to fill a selected volume (1.15 L) of the experimental column. The weight of each substrate was measured in triplicate and exposed to untreated AMD in triplicate columns. Untreated AMD (2 L) was added to each substrate and sealed for static exposure at room temperature for a period of 35 days. (Table 1 lists the different weights of substrates used to fill 1.15 L of the experimental column and Figure 4 provides a visual representation of the phase 1 media composition.) A blank sample was added, which only contained filter media in the experimental triplicate columns and ±3 L untreated AMD water during Phase 1. This was added to see the impact of time and filter media on the static column.

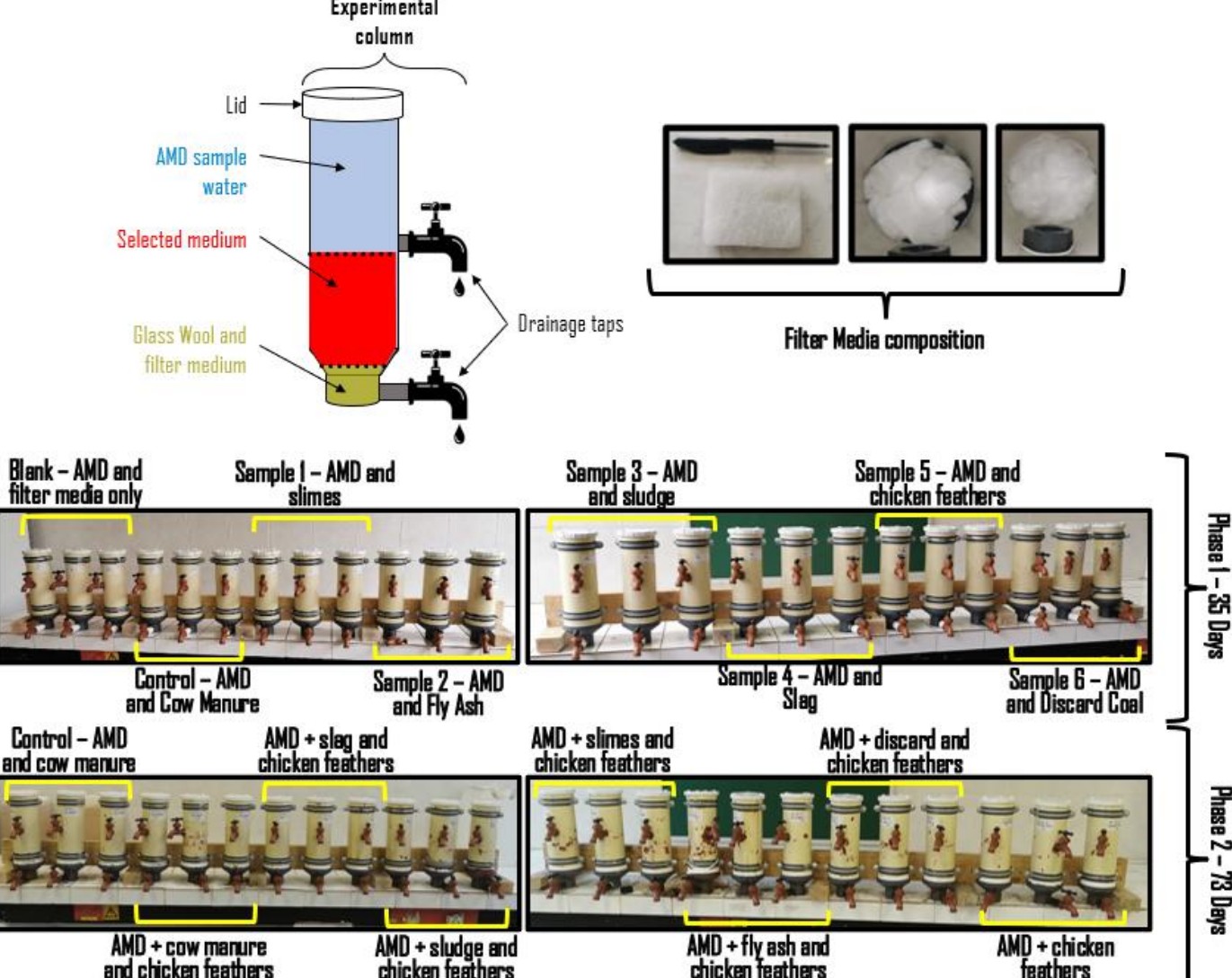

**Figure 3.** Typical experimental column composition and actual experimental set-up for both Phase 1 and 2.

**Table 1.** Details of substrates prepared for various experimental columns at Phases 1 and 2.

| PHASE 1: AMD Volume ± 2 L | | | | PHASE 2: AMD Water Volume Was 2.5 L Substrate Volume Comprised 2/3 Substrate and 1/3 CF | | | |
|---|---|---|---|---|---|---|---|
| Substrate | Volume | Comments | Total Weight | Substrate | Volume | Comments | Total Weight |
| CM | | Used as control | 540 g | CM | | 190 g CM—Control | 190 g |
| Slimes | | Experimental substrate— aimed to understand what the various media do to the AMD over time. | 940 g | Slimes & CF | | 230 g Slimes + 26 g CF | 256 g |
| FA | | | 1250 g | FA & CF | | 310 g FA + 26 g CF | 336 g |
| Sludge | 1.15 L | | 1820 g | Sludge & CF | 0.4 L | 480 g Sludge + 26 g CF | 506 g |
| Slag | | | 2200 g | Slag & CF | | 630 g Slag + 26 g CF | 656 g |
| Discard | | | 1400 g | Discard & CF | | 360 g Discard + 26 g CF | 386 g |
| CF | | | 200 g | CF | | 120 g CF | 120 g |
| Blank | 3 L | No substrate, only filter media and AMD | N/A | CM & CF | | 130 g CM + 26 g CF | 156 g |

**Filter Media—Cumulative weight for each experimental column was 16 g (not added to total reported weight)**

Note: AMD = acid mine drainage; CF = chicken feathers; CM = cow manure; FA = fly ash; N/A = not applicable.

Phase 1

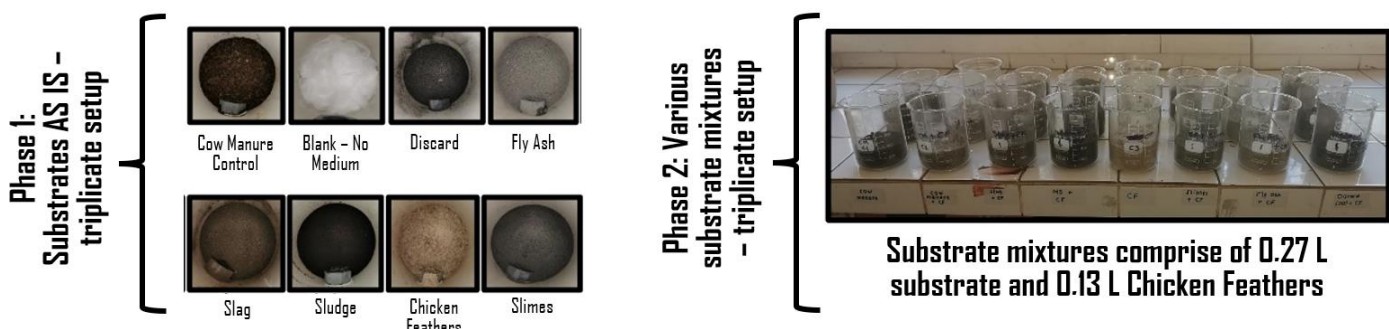

**Figure 4.** Visual representation of various media compositions as prepared for Phase 1 (35 days) and Phase 2 (73 days) exposure to AMD.

Phase 2

During Phase 2, the substrate that performed optimally (i.e., chicken feathers) was mixed with the other substrates tested in Phase 1. The objective was to evaluate if it would be possible to enhance the organic carbon content of the various substrates exposed to AMD water during Phase 1 and achieve improved AMD water remediation, i.e., improved water quality. The various substrates were prepared as mixtures with chicken feathers in triplicate. A constant weight of chicken feathers (26 g) (volume level of 130 mL) was individually added to each substrate. The quantity and weight of the chicken feathers were kept constant throughout all the substrate mixtures. The theoretic approach aimed to determine the efficacy of these substrates if one cup of each were added to untreated AMD. (Figure 4 depicts a visual representation of the phase 2 media.) The various substrate volumes (in terms of the space each substrate took up) were also kept constant for the mixed substrate preparations (267 mL), but, due to density differences between the various substrates, the weights differed. The untreated AMD volume added to the different substrates was constant at 2.5 L. After the addition of the AMD to the substrate mixture, the columns were sealed for static exposure at room temperature for a period of 73 days.

*2.3. Chemical/Biological Analyses*

2.3.1. Substrates

Pre-AMD Exposure (Day O)

The various substrate samples used in both Phase 1 and 2 of the experiment were collected pre-AMD exposure (Day 0) and applied for chemical analyses (Table 2). For Phase 1, a total of seven substrates (non-triplicate) were analyzed, and in Phase 2 a total of eight substrate mixtures were applied for analyses pre-AMD exposure. The analyses completed for Phase 1 included total organic carbon (TOC), total carbon (TC) and total inorganic carbon (TIC) and was required to understand the carbon composition of the selected substrate materials. The specific element of interest was TOC, as it is known to support microbial activity for $SO_4$ reduction in AMD [6,7,11,13,16,19,22–24]. Phase 2 substrate mixtures were also analyzed for TOC, TC and TIC; however, additional analyses were requested for the substrate mixtures during Phase 2. These additional analyses included nitrogen (N), phosphorous (P) and water-soluble P (Table 2) and were identified as potential electron donors for microbial activity.

**Table 2.** Test methods applied to substrate analyses.

| Selected Chemical Parameters | Analyses Method |
|---|---|
| Total carbon % | Medium combustion and infrared detection (LECO CS23o instrument) |
| Total organic carbon (%) | Boiling medium in diluted acid (removing inorganic carbon as $CO_2$); remaining organic carbon is dried and analysed by combustion and infrared detection (LECO CS23o instrument) |
| Total inorganic carbon (%) | Calculated from total carbon and organic carbon results (TC − TOC = TIC) |
| Total nitrogen (%) | DUMAS/Combustion method |
| Total phosphorous (%) | Inductively Coupled Plasma Optical Emission spectroscopy (ICP-OES) |
| Water-soluble phosphorous (%) | Inductively Coupled Plasma Optical Emission spectroscopy (ICP-OES) |

Post-AMD Exposure (Day 35—Phase 1; Day 73—Phase 2)

The triplicate samples exposed to AMD were collected and sent for analyses during Phase 2 only. This was identified as a requirement from Phase 1. The analyses concluded on the 24 samples (eight substrates exposed in triplicate) collected post-AMD exposure, included TC, TOC, TIC, N, P and water-soluble P, and aimed to provide insight into the specific substrate mixtures' ability to support microbial activity.

Test Methods for Substrate Analyses

Table 2 summarises the various test methods used for the analyses of the selected parameters on the various substrate samples that were submitted.

2.3.2. Water Samples

Untreated AMD (Day 0)

The pre-treatment AMD water samples collected from the reclamation plant were sent for chemical analyses (Table 3). The pre-treatment AMD water was collected in triplicate (1 L each) for both phases. The analysed parameters included total organic carbon (TOC), pH, total dissolved solids (TDS), electrical conductivity (EC), alkalinity (Alk), ammonia and ammonium ($NH_3$ and $NH_4$), nitrate ($NO_3$), nitrite ($NO_2$), orthophosphate ($PO_4$), sulphate ($SO_4$), aluminium (Al), iron (Fe), manganese (Mn) and sulphide ($S_2$).

**Table 3.** Test methods applied to water sample analyses.

| Selected Chemical Parameters | Analyses Method |
| --- | --- |
| Acidity (mg CaCO$_3$/L) | Titrimetric (indicator/pH): Acidity |
| Total alkalinity (mg CaCO$_3$/L) | Titrimetric (indicator/pH): Alkalinity |
| pH | pH: Electrometric |
| Oxidation reduction potential (mV) | mV Reading: Electrometric |
| Total dissolved solids (mg/L) | Calculation: Adding cations and anions measured |
| Ammonium and ammonia (mg N/L) | Spectrometry: Ammonia calculated according to pH, ammonium and ammonia concentration |
| Total organic nitrogen (mg N/L) | Spectrometry calculation |
| Nitrite (mg N/L) | Spectrometry |
| Orthophosphate (mg P/L) | Spectrometry |
| Sulphate (mg SO$_4$/L) | Spectrometry |
| Aluminium (mg Al/L) | ICP Spectrometry |
| Iron (mg Fe/L) | ICP Spectrometry |
| Manganese (mg Mn/L) | ICP Spectrometry |
| Total organic carbon (mg C/L) | TOC analyser: Analytiklena combustion |
| Sulphide (mg S$_2$/L) | Titrimetric (combined with metals) |

Remediated AMD (Day 35—Phase 1; Day 73—Phase 2)

The triplicate water samples exposed to the various substrates of the different phases were collected (±1 L each) and sent for chemical analyses. The same water parameters were analysed as per the untreated AMD water samples for both phases. For Phase 1, a total of 24 samples (seven substrate-exposed AMD water samples in triplicate and one set of triplicate blank samples) were sent for analyses. Phase 2 also had a total of 24 samples (eight mixed substrate-exposed AMD water samples in triplicate) submitted for chemical analyses.

Test Methods for Water Analyses

Table 3 summarises the various test methods used for the analyses of the selected parameters of the submitted water samples.

2.3.3. Microbial Diversity—16 S rRNA Sequencing

The microbial diversity involved in the AMD treatment process was determined using 16S rRNA next-generation sequencing according to the methods described by Azaroual et al. [25], Chen et al. [26] and ThermoFisher [27]. The sequencing was performed by the Department of Genetics, Faculty of AgriSciences, University of Stellenbosch, Western Cape, South Africa. Only remediated AMD water samples originating from Phase 2 were studied using 16S rRNA sequencing. This was used to evaluate the selected substrate mixtures' potential to support the microbial biodiversity required for SO$_4$ reduction and AMD water remediation. A total of 25 samples (one set of AMD water representing Day 0 and 24 (eight triplicate samples) remediated AMD water samples collected on Day 73) were subjected to 16S rRNA sequencing. The collected water samples, AMD (Day 0) and remediated AMD (Day 73), were kept at room temperature during transportation to the laboratory for analyses. The samples were subjected to the culture-free sequencing method using the Ion 16S Metagenomics Kit (Thermofisher Scientific, Waltham, MA, USA).

The Ion Reporter software uses a set of algorithms to analyse the sequenced data and cluster the 16S rRNA sequencing, based on similarities between sequences, into the various Operational Taxonomic Units (OTUs). These were used to estimate the richness, composition, and diversity of the microbial populations in the samples [26]. The results were analysed and grouped into OTUs whose numbers varied throughout the taxonomic classification system, delivering results detailing the order level in most cases, followed by the family level, and, in some instances, genera levels were represented. These data were further simplified and grouped into specific families according to the identified OTUs.

*2.4. Data Analyses*

This study aimed to explore alternative substrates and their potential to initiate the passive remediation of AMD. The research design is of a quasi-experimental nature, which assisted in determining if there was a change from the situation before treatment versus the situation after treatment. The comparisons were concluded based on the volumes of exposed substrates. The obtained data were subjected to a comparative analysis to determine if AMD water quality improved after exposure to the various substrate volumes. The various substrate performances were measured against that of the control (cow manure). The substrate analyses completed for each phase prior to AMD exposure were obtained from a single analysis. Analyses completed on the substrate after exposure to AMD were completed in triplicate. The substrate weight per volume occupied within the column was also considered to verify the relevant treatment capacity of the specific substrates relative to their weight and density. The water chemistry of both the AMD and remediated AMD samples were also sampled in triplicate.

Variance in water quality parameters were assessed using mixed models with substrate as a fixed factor and replicate as a random factor (VEPAC Module, Statistica V14, Tibco Inc., Palo Alto, CA, USA). The normality of the data was assessed using normal probability plots of residuals. Data that were not normally distributed were ranked or boxcox, transformed prior to analysis. Water chemistry values below the detection limit of the instruments applied were considered as zero. Pairwise differences were assessed using Fisher's LSD post hoc test. A *p*-value of <0.05 was considered significant.

A principal component analysis (PCA) was performed to assess the associations between selected water quality variables and the microbial taxa present in the different substrate types represented in Phase 2 of the experiment. A PCA tri-plot was produced using Canoco V5 (Microcomputer Power, Ithaca, NY, USA) and featured microbial data as a focal plot and water chemistry as supplementary variables. The microbial data were log-transformed, centred, and standardized prior to the PCA. Mean values (representing three replicates) were applied to water chemistry variable per substrate type.

## 3. Results and Discussion

Passive treatment technologies are reliant on natural processes, including plant-mediated and microbial activities, as well as chemical and physical processes, to treat AMD [12]. To support the microbial activity that is responsible for AMD treatment, organic carbon content is essential. SRB have a carbon preference and require organic electron donors to drive $SO_4$ reduction processes. Various organic materials may present these organic electron donors; however, their availability is dependent on the chemical characteristics and degradation method of the specific organic material [5]. Protein is also listed as an organic electron donor, with 50–55% of carbon as part of its typical composition. Organic carbon availability to SRB allows for $SO_4$ reductions under anaerobic conditions. The selected substrates provide attachment sites for microorganisms that perform essential functions, as well as metals and nutrient sorption sites [16,19].

In constructed wetlands, waste substrates, including cow, chicken and sheep manure, wood chips, sawdust, peat, hay, sewage sludge, methanol, ethanol, compost, fly ash, slag, zerovalent iron particles and kiln waste, have been applied to treat AMD successfully and efficiently [10,19]. A diverse range of organic substrates have been studied and cow manure mixtures (mixed with alternative organic media such as straw and hay) have provided a good treatment capacity within passive treatment systems [19]. Choudhary and Sheoran [28] found that single substrate cow manure increases AMD pH from 2.70 to 6.25 and effectively removes metals from AMD within 10 days of exposure, while cow manure mixtures managed to elevate the pH from 2.1 to 6.4, achieving successful metal removal (apart from manganese) and sulphate reduction rates of 44–75% [29]. Cow manure is, therefore, considered a good organic substrate that can support the microbial activity responsible for AMD treatment in passive treatment systems, and was used as a control sample in both phases of this experiment. The selection of the other alternative substrates

was based on their relative abundance, availability, low cost, organic carbon, and waste status, driving the need for a circular economy. Although two of the substrates that were selected, fly ash and slag, have been subjected to previous studies for their effectiveness in treating AMD, these were not subjected to enhancement mixtures in the literature.

### 3.1. Phase 1 and 2 Observations

The Phase 1 experimental column set-up aimed to determine the passive impact that the volume of the selected substrates will have on AMD during a static exposure of 35 days. The Phase 2 experimental column set-up aimed to determine the passive impact of the various selected substrate chicken feathers' mixed volumes on AMD during a static exposure of 73 days. The analyses of the substrates made prior to AMD exposure aimed to determine the carbon content and possible organic carbon presences that were capable of supporting SRB activity during the static exposure timelines. The substrates were not retrieved post-AMD exposure for analyses during Phase 1. Analyses of the substrates post-AMD exposure were identified as a requirement for Phase 2, to assist in determining the extent to which the various substrates' available nutrients and electron donors may support SRB growth during static exposure to AMD. Chicken feathers are considered a high-quality protein supplement, consisting of more than 85% protein [30]. Additional analyses were selected for the Phase 2 substrate mixtures because of the high protein content of chicken feathers. These included P, N and water-soluble P. The TOC (both phases), N, P and water-soluble P (Phase 2 only) would provide an indication of the potential available (pre-AMD exposure) and used (post-AMD exposure) organic electron donors for the SRB in the various substrates mixtures to enable the $SO_4$ reduction process within the AMD.

From visual observations, the chicken feathers showed hydrophobic characteristics during the column preparation period for Phase 1, resulting in buoyancy for the chicken feather substrates. Cow manure showed buoyancy properties in both phases. A distinct black colour change was noted in the AMD water collected from the Phase 1, 35-day static exposure experiment with the cow manure and chicken feathers substrates was noted. A rotten egg smell was also observed, indicative of SRB activity releasing $H_2S$ gas from the $SO_4$ reduction process. The collection of the exposed AMD water after the 73-day static exposure timeframe for Phase 2 revealed differences in the colour of the samples collected from the experimental columns. An approximate volume of two litres was collected from each experimental column that was prepared for Phase 2 exposure after the 73-day exposure timeframe lapsed. The first one-litre exposed AMD volume collected displayed a definite darker colour compared to the second 1-L exposed AMD volume collected from the same experimental column. This was true for all but one sample triplicate substrate mixture—the sludge and chicken feathers substrate mixture—which revealed AMD-exposed collected water samples with no variation in colour. The smell of rotten egg was evident in all water samples collected after the 73-day exposure timeframe for Phase 2, indicative of $H_2S$ gas formation. It was expected that the AMD saturated substrate mixtures in the experimental columns created anaerobic conditions associated with the water-infiltrated media, compared to the less anaerobic conditions expected in the water column on top of the media. This may have contributed to the observations of darker coloured water being collected in the first 1-L sample that was drained from the experimental column, since SRB activity may have been promoted in the more anoxic layer of the experimental column. The water samples were mixed after collection from the experimental columns to create a homogenous mixture, which was submitted for water and microbial analyses.

### 3.1.1. Substrates
Chemical Results

The results obtained for Phase 1 and 2 substrates are summarised in Table 4. The pre-AMD exposure analyses completed for both experimental phases are reported as a single analysis (pre-AMD exposure substrates were not analysed in triplicate). The post-AMD exposure results for Phase 2 substrates are reported as the average percentage of the

triplicate analyses completed for each mixed substrate. The standard deviations for these triplicate analyses are also listed.

Discussion—Phase 1 Substrate Results

The Phase 1 substrate results, pre-AMD exposure, confirmed the presence of carbon in all selected substrates. The substrate with the highest carbon content was chicken feathers, with 46.7%. However, this substrate also showed the lowest TOC content of 2.36% (Table 4). Considering that chicken feathers are composed of keratin (protein), this carbon is captured in the protein structure and requires hydrolyses. The hydrolysis of proteins releases amino acids, the smaller building blocks of proteins, which are metabolised by microorganisms to release organic acids, alcohols, carbon dioxide and ammonia [3,30]. The cow manure substrate had a 12.2% TOC content, which was exceeded by discard at 16.9%. The TOC concentration of slimes was the third highest at 9.4%. The slag substrate had the lowest TOC content at 0.93% (13 times lower than that of cow manure). Fly ash, sludge and chicken feathers displayed three to four times lower TOC content compared to the cow manure, at 3.69%, 2.52% and 2.36% each, respectively (Table 4). The TOC content in the remediated AMD water after exposure (Day 35) was high for the chicken feathers with remediated AMD. Based on the high TOC content seen in the chicken feathers in the remediated AMD water at Day 35, Phase 2 was designed to add a constant volume of chicken feathers to a constant volume of each substrate to enhance TOC availability. The addition of chicken feathers to the various substrates aimed to enrich the organic composition of the Phase 2 substrates, with the possibility of enhancing the microbial growth and remediation achieved in Phase 1, potentially resulting in an improved AMD treatment capacity for the mixed substrates.

Discussion—Phase 2 Substrate Results

During Phase 2, the single-phased substrates were mixed with chicken feathers to see if the protein would enhance and possibly better support the microbial activity in the experimental columns within the 73-day exposure timeframe. Cow manure was again used as a single-phase substrate as the control in Phase 2 to measure the performance of the other mixed substrates that were exposed during Phase 2. The Phase 2 mixed-substrate samples submitted for analyses for both pre- and post-AMD exposure during the Phase 2 static experiment, were single and triplicate samples, respectively, as summarised in Table 4. The standard deviations in the TOC results from the triplicate post-AMD exposed substrates indicate higher values for the chicken feathers, cow manure and the substrate mixture of cow manure and chicken feathers (5.5, 11.6 and 13.7, respectively). This was indicative of a larger variation in these substrates' overall TOC content, with the larger variations being associated with the cow-manure-containing substrates. This variation may be attributed to the sampling procedure and substrate preparation of the cow manure, as multiple rocks or stones formed part of the cow manure sample, which would add variations to the organic nature of this substrate content. From the additional analyses completed on the Phase 2 substrates (N, P and water-soluble P), the N content may have contributed to the support of SRB as an electron donor in the sulphate reduction process [31]. However, this was to a much smaller extent, which was observed for the TOC availability. Although P and water-soluble P were also detected in selected substrate mixtures, these were detected at very low concentrations and were not seen as a major contributing factor to the $SO_4$ reduction process associated with SRB.

**Table 4.** Substrate analyses pre- and post-AMD exposure for both experimental phases. Values represent mean ± standard deviation.

| Phase 1—Single Substrate | | CF | CM | Slag | Sludge | Slimes | FA | Discard | |
|---|---|---|---|---|---|---|---|---|---|
| **Pre-AMD exposure** | % TC | 46.7 | 22.6 | 1.57 | 8.73 | 11.3 | 18.1 | 18.1 | |
| | % TOC | 2.36 | 12.2 | 0.93 | 2.52 | 9.40 | 16.9 | 16.9 | |
| | % TIC | 44.3 | 10.4 | 0.64 | 6.21 | 1.90 | 1.20 | 1.20 | |
| **Phase 2—Mixed substrates** | | CF Alone | CM Alone | Slag and CF | Sludge and CF | Slimes and CF | FA and CF | Discard and CF | CM and CF |
| **Pre-AMD exposure** | % TC | 48.66 | 18.63 | 3.27 | 8.65 | 15.64 | 23.15 | 11.86 | 24.48 |
| | % TOC | 48.65 | 18.56 | 3.16 | 8.65 | 14.64 | 22.27 | 11.72 | 24.41 |
| | % TIC | 0.01 | 0.01 | 0.12 | 0.05 | 1.0 | 0.88 | 0.14 | 0.07 |
| | % N | 14.8 | 1.75 | 0.77 | 0.7 | 1.38 | 0.12 | 1.23 | 3.63 |
| | % P | 0.1 | 0.34 | ND | 0.02 | 0.09 | 0.07 | 0.04 | 0.30 |
| | % Water-soluble P | 0.02 | 0.07 | ND | ND | ND | ND | ND | 0.06 |
| **Phase 2—Mixed substrates** | | CF Alone | CM Alone | Slag and CF | Sludge and CF | Slimes and CF | FA and CF | Discard and CF | CM and CF |
| **Post-AMD exposure** | % TC | 37.8 ± 5.5 | 26.9 ± 11.6 | 2.8 ± 0.38 | 9.3 ± 1.81 | 13.7 ± 0.51 | 5.1 ± 0.36 | 17.1 ± 0.85 | 16.1 ± 13.7 |
| | % TOC | 37.8 ± 5.5 | 26.9 ± 11.6 | 2.7 ± 0.35 | 9.2 ± 1.80 | 12.5 ± 0.49 | 4.2 ± 0.30 | 17.0 ± 0.86 | 16.1 ± 13.7 |
| | % TIC | 0 ± 0.01 | 0.02 ± 0.01 | 0.2 ± 0.03 | 0.1 ± 0.03 | 1.2 ± 0.04 | 0.9 ± 0.06 | 0.1 ± 0.01 | 0.03 ± 0.01 |
| | % N | 10.8 ± 1.6 | 2.06 ± 1.7 | 0.41 ± 0.11 | 0.9 ± 0.15 | 1.2 ± 1.01 | 1.0 ± 0.10 | 1.2 ± 0.13 | 2.2 ± 1.58 |
| | % P | 0.07 ± 0.01 | 0.20 ± 0.03 | 0.01 ± 0.01 | 0.02 ± 0.00 | 0.06 ± 0.01 | 0.07 ± 0.00 | 0.04 ± 0.00 | 0.20 ± 0.04 |
| | % Water-soluble P | ND | 0.01 ± 0.00 | ND | ND | ND | ND | ND | 0.01 ± 0.00 |

Note: AMD = acid mine drainage; CF = chicken feathers; CM = cow manure; FA = fly ash; ND = no data; TC = total carbon; TOC = total organic carbon; TIC = total inorganic carbon.

3.1.2. Water Samples

Chemistry Results—Phase 1

　　The water chemistry results of the AMD samples pre- and post-substrate exposure (AMD and remediated AMD, respectively) during Phase 1 are summarised in Table 5. There was no significant difference between the blank AMD sample (exposed to 35-day static timeframe with no substrate exposure) and the AMD water (Day 0). The various *p*-values for pH, $SO_4$, Fe and alkalinity was 0.093, 0.869, 0.339 and 1, respectively, indicative of there being no significant change in the water chemistry of the AMD related to the exposed time period. It can be concluded that chemical parameter changes in the water column of the other static substrate-exposed AMD (remediated AMD), resulted from the chemical or microbial interaction with the substrate. Comparing each substrate's remediated AMD $SO_4$ content to that of the control (cow manure) provides insight into each substrate's ability and performance in remediating AMD.

**Table 5.** AMD water (pre-exposure and remediated AMD) chemical analysis—Phase 1: Reported as the mean.

| Parameter | Pre-Exposure AMD | Remediated AMD | | | | | | | |
|---|---|---|---|---|---|---|---|---|---|
| | | **Blank** | **CM** | **Slimes** | **FA** | **Sludge** | **Slag** | **CF** | **Discard** |
| Alkalinity (mg $CaCO_3$/L) | 0 | 0 | 6 313 | 217 | 283 | 6 | 27 | 2340 | 324 |
| pH | 2.62 | 2.54 | 7.04 | 7.55 | 6.52 | 4.70 | 5.43 | 6.01 | 6.37 |
| TDS (mg/L) | 4 533 | 4521 | 14,752 | 13,574 | 5333 | 3272 | 3836 | 6530 | 7656 |
| $SO_4$ (mg/L) | 3 325 | 3357 | 1939 | 6997 | 2934 | 2387 | 2770 | 2996 | 5344 |
| Al (mg/L) | 72 | 85 | 1 | 0 | <0.01 | 0.85 | 5 | <0.01 | 0 |
| Fe (mg/L) | 264 | 227 | 0 | 0 | 1 | 13 | 5 | 1 | 0 |
| Mn (mg/L) | 26 | 26 | 4 | 5 | 16 | 24 | 28 | 19 | 19 |
| $NH_3$ and $NH_4$ (mg N/L) | 9 | 7 | 332 | 5 | 9 | 11 | 4 | 879 | 3 |
| $NO_2$ and $NO_3$ (mg N/L) | <0.35 | <0.35 | <0.35 | 48 | 22 | <0.35 | <0.35 | <0.35 | <0.35 |
| $NO_2$ (mg N/L) | <0.01 | <0.01 | <0.01 | 20 | 2 | <0.01 | <0.01 | <0.01 | <0.01 |
| Ortho-P (mg P/L) | <0.03 | 0.065 | 50 | 0.057 | <0.03 | <0.03 | <0.03 | 1.05 | <0.03 |
| $S_2$ (mg/L) | 3 | 5 | 71 | <0.5 | 1 | 7 | 4 | 16 | 5 |
| TOC (mg C/L) | 3 | 19 | 4219 | 19 | 78 | 74 | 25 | 2409 | 48 |

Note: AMD = acid mine drainage; CF = chicken feathers; CM = cow manure; FA = fly ash; SD = standard deviation.

　　In Phase 1, passive treatment was initiated in the various static experimental columns during the 35-day exposure time. In Phase 2, passive treatment of the remediated AMD water samples (73-day exposure) was evident from the pH elevation and $SO_4$ reduction noted in all remediated AMD samples compared to the AMD water sample (Day 0). Studies have shown that the optimum pH for SRB activity falls within the ranges of 5–8 [13]. During Phase 1, the pH of the remediated AMD water changed from 2.6 for the AMD to ranges between 4.7 (sludge) and 7.6 (slimes) for the remediated AMD after exposure to different substrates. The statistical data indicated the pH changes were significant, with *p*-values well below 0.05 for cow-manure-, discard-, chicken-feathers-, fly-ash-, slimes- and sludge-remediated AMD. Metal removal, specifically Fe content, was significant, with *p*-values well below 0.05 for all substrate-remediated AMD. $SO_4$ decreases were evident in the range from 10% to 43% in the remediated AMD. The changes noted in the slimes and fly ash samples were not seen as significant, with *p*-values of 0.152 and 0.063, respectively. Although five of the seven remediated AMD water samples indicated a reduction in $SO_4$ content in the AMD, only four of these had *p*-values indicating significant differences in $SO_4$ content, compared to the original AMD (pre-treatment) $SO_4$ water sample. These were the slag, sludge, discard and cow manure, and the observed $SO_4$ changes. The reduction achieved in the fly ash remediation AMD was not significant (*p*-value of 0.063). Two remediated AMD samples (slimes and discard) showed an increased $SO_4$ content of 110% and 60%, respectively, compared to the AMD water chemistry. The degree of Fe, Al, Mn, $SO_4$ and

TOC concentration changes noted within the various remediated AMD water samples (Day 35), compared to the AMD water sample (Day 0) is summarised in Figure 5.

**Figure 5.** Percentage change in selected chemical parameters and pH observed comparing AMD (Day 0) with remediated AMD water (Day 35).

The cow manure substrate achieved the highest $SO_4$ reduction rate (42%) in the remediated AMD water (Day 35) compared to the AMD water (Day 0), followed by the $SO_4$ reduction in the sludge (28%), slag (17%), fly ash (12%) and chicken feathers (10%) substrate-remediated AMD water (Day 35). These changes were observed relative to the substrate volume used. Standardising the $SO_4$ reduction rate against the mass of each substrate relative to the mass of the cow manure control meant that the $SO_4$ reduction performance per substrate could be recalculated as 27% for chicken feathers, 42% for cow manure (remains the same), 4% for slag, 8% for sludge and 5% for fly ash. Table 6 summarises this relative $SO_4$ weight reduction potential compared to the assessed volume reduction potential.

**Table 6.** Phase 1 $SO_4$ reduction rate calculated relative to cow manure control sample weight.

| $SO_4$ Reduction—Phase 1 | CM | Sludge | Slag | FA | CF | Slimes | Discard |
|---|---|---|---|---|---|---|---|
| **Volume-related reduction achieved (1.15 L substrate)** | 42% | 28% | 17% | 12% | 10% | 110% increase | 60% increase |
| **Weight-related reduction achieved relative to cow manure carbon content analysed** | 42% | 8% | 4% | 5% | 27% | 63% increase | 23% increase |

Note: CM = cow manure; FA = fly ash; CF = chicken feathers.

Obtaining optimal conditions within passive treatment systems to sustain microbial populations is essential, and the degradation of the organic carbon source by SRB in such systems may also release residual organics and nutrients, which could further support SRB growth [3]. TOC in the remediated AMD water samples was highly elevated in the chicken feathers substrate-remediated AMD water (Day 35) at 2419 mg/L, supported by a *p*-value of <0.001, indicating significance. The remaining substrate's remediated AMD water (Day 35) had much lower levels of TOC presence (ranges from 15.6 mg/L to 92 mg/L),

yet all *p*-values indicated significant changes. The TOC of the cow-manure-remediated AMD water (Day 35) almost doubled, at 4 209 mg/L. The elevation in the TOC parameter of chicken-feathers-remediated AMD water may be a good electron donor for microbial activity within the substrate. Zhang and Wang [31] referred to the N content of SRB-supporting media as an important factor which impacts the performance of AMD treatment by SRB. The hydrolysis of proteins release amino acids, the smaller building blocks of proteins, which are metabolized by microorganisms to release organic acids, alcohols, carbon dioxide and ammonia [3]. Nitrogen was detected in the form of ammonia/ammonium at elevated levels in the remediated AMD water of the chicken feathers substrate at 879 mg/L (2.6 times higher than that detected in the cow-manure-remediated AMD water). The chicken feathers substrate showed great potential to deliver available electron donors to increase microbial activity in the water column, as is evident from the elevated TOC and nitrogen levels in the remediated AMD. For Phase 2, the various substrates were mixed with the chicken feathers substrate to explore the performance changes that may result in the passively treated AMD exposed to the chicken feather enhanced substrate mixtures.

Chemistry Results—Phase 2

The chemistry results of the AMD water samples pre- and post-substrate exposure (AMD water at Day 0 and remediated AMD water at Day 73, respectively) during Phase 2 are captured in Table 7. Elevated pH levels were achieved in all remediated AMD water samples (Day 73) during the Phase 2 static exposure timeframe. The remediated AMD water pH ranges (Day 73), in comparison with the pre-exposure AMD water (Day 0), were between 5.73 (sludge and chicken feathers) and 7.87 (slimes and chicken feathers). All of these were significant changes, with very low calculated *p*-values. The slimes- and chicken-feathers-, and fly-ash- and chicken-feathers-remediated AMD water samples (Day 73) were the only two remediated AMD water samples with a pH above that of the cow manure control (7.87 and 7.32, respectively, versus 7.20). These variations were indicated to be significant for cow manure, with *p*-values of 0.004 and <0.001, respectively. Alkaline pH levels in the remediated AMD water samples (Day 73) create precipitating conditions for dissolved metals, forming metal sulphides, -oxides, -hydroxides and -carbonates [13]. Alkalinity increased in all substrate mixes, with the control remediated AMD achieving the highest alkalinity presence (2454 mg/L). The cow manure and chicken feathers mixtures' remediated AMD water samples (Day 73) achieved very similar alkalinity results to that of the control (2450 mg/L). All alkalinity changes in the remediated AMD of the various substrates were indicated to be significant changes in the statistical analysis. The pH and corresponding metal and $SO_4$ reductions achieved by the remediated AMD water samples (Day 73), in comparison to the relevant AMD water pre-substrate-exposure (Day 0), are captured in Figure 6.

The $SO_4$ reduction rate of the cow manure control (comparing Day 0 AMD $SO_4$ content with Day 73 remediated AMD $SO_4$ content) for Phase 2 was 66% ($p < 0.001$). The $SO_4$ reduction in the cow manure and chicken feathers mixtures' remediated AMD water (Day 73) was very similar to the control, at 62% with a significant *p*-value. The fly ash and chicken feathers remediated AMD water samples (Day 73) did show a $SO_4$ reduction rate of 50%. Overall, a $SO_4$ reduction was achieved in all the substrate mixtures' remediated AMD water; however, all $SO_4$ reduction rates were lower compared to the control. These changes were relative to the substrate volume that was used. By standardising the $SO_4$ reduction rate against the mass of each substrate relative to the mass of the cow manure control, the $SO_4$ reduction performance per substrate could be recalculated as 63% for chicken feathers, 66% for cow manure (remains the same), 9% for slag and chicken feathers, 13% for sludge and chicken feathers, 29% for slimes and chicken feathers, 28% for fly ash and chicken feathers and 11% for discard and chicken feathers. Table 8 summarises this relative $SO_4$ weight reduction potential compared to the assessed volume reduction potential.

**Table 7.** AMD water (pre-exposure and remediated AMD) chemical analyses—Phase 2: Reported as mean).

| Parameter | Pre-Exposure (Day 0) | Remediated AMD Water (Day 73) | | | | | | | |
|---|---|---|---|---|---|---|---|---|---|
| | AMD | CM & CF | CM | Slimes & CF | FA & CF | Sludge & CF | Slag & CF | CF | Discard & CF |
| Alkalinity (mg/L) | 0 | 2450 | 2454 | 1542 | 1819 | 101 | 390 | 839 | 559 |
| pH | 3.00 | 7.03 | 7.20 | 7.87 | 7.32 | 5.73 | 7.10 | 6.81 | 6.16 |
| TDS (mg/L) | 4 351 | 5131 | 5432 | 4404 | 4277 | 2938 | 3428 | 3436 | 3998 |
| SO$_4$ (mg/L) | 3 208 | 1218 | 1085 | 1843 | 1605 | 2071 | 2250 | 1919 | 2497 |
| Al (mg/L) | 68 | 0.04 | 0.22 | 0.01 | 0.01 | 0.01 | 0.01 | 0.01 | 0.01 |
| Fe (mg/L) | 245 | 0.17 | 0.26 | 0.39 | 0.13 | 31 | 0.34 | 0.19 | 17 |
| Mn (mg/L) | 25.4 | 9.5 | 7.4 | 2.2 | 1.3 | 16.6 | 17.4 | 16.6 | 19.8 |
| NH$_3$ & NH$_4$ mg/L | 12 | 146 | 49 | 89 | 201 | 61 | 65 | 153 | 48 |
| Total N (mg/L) | 12 | 146 | 49 | 89 | 201 | 61 | 65 | 153 | 48 |
| Ortho-P (mg/L) | <0.03 | 25 | 34 | 0.12 | 0.93 | <0.03 | <0.03 | 0.06 | <0.03 |
| S$_2$ (mg/L) | <0.50 | <0.50 | <0.50 | <0.50 | <0.50 | <0.50 | <0.50 | <0.50 | <0.50 |
| TOC (mg C/L) | 20 | 1011 | 361 | 464 | 1543 | 142 | 26 | 1040 | 85 |

Note: AMD = acid mine drainage; CF = chicken feathers; C = cow manure; FA = fly ash.

**Figure 6.** Percentage changes in selected chemical parameters and pH observed comparing AMD (Day 0) with remediated AMD water (Day 73).

**Table 8.** Phase 2 SO$_4$ reduction rate calculated relative to cow manure control sample weight.

| SO$_4$ Reduction—Phase 2 | CM | Sludge and CF | Slag and CF | FA and CF | CF | Slimes and CF | Discardand CF |
|---|---|---|---|---|---|---|---|
| Volume-related reduction achieved (1.04 L substrate) | 66% | 35% | 30% | 50% | 40% | 43% | 22% |
| Weight-related reduction achieved relative to cow manure carbon content analysed | 66% | 13% | 9% | 28% | 63% | 29% | 11% |

Note: CM = cow manure; FA = fly ash; CF = chicken feathers.

Iron reduction rates in the remediated AMD water (Day 73) of 100% (same as in the control cow manure) were achieved in the cow-manure- and chicken-feathers-, slimes- and

chicken-feathers-, fly-ash and chicken-feathers-, slag- and chicken-feathers-, and chicken-feathers-substrate-remediated AMD water (Day 73). The iron reduction rate for the sludge- and chicken-feathers-, and discard- and chicken-feathers-remediated AMD water (Day 73) was achieved at 87% and 93%, respectively. The *p*-values calculated in relation to Fe content changes were all indicative of being significant based on the volume of the substrate treatment.

Elevated ammonia concentrations were detected in all remediated AMD water samples and are an indication of microbial activity [3]. The highest level of ammonia presence was detected in the fly ash and chicken feathers substrate (201 mg/L) (4.1 times higher compared to the control), with a $p < 0.001$. The lowest ammonia levels were detected in the discard and chicken feathers substrate (48 mg/L) and a resultant *p*-value of 0.059 (non-significant). Although some colour variations were observed in the remediated AMD water samples (Day 73), accompanied by a rotten egg odour, the chemical analyses did not detect sulphide concentrations above 0.5 mg/L in any of the remediated AMD samples. The percentage availability of nutrients (TOC, P and N) calculated from the substrate mixtures in the remediated AMD water samples (Day 73) revealed that the highest TOC and N percentages (calculated as the sum of $NH_3$ and $NH_4$, $NO_2$ and $NO_3$) detected in the remediated AMD (Day 73) were detected in the fly ash and chicken feathers substrate (Table 9). Elevated phosphorous concentrations in the remediated AMD water (Day 73) were mostly detected in the cow-manure-containing substrates. The nutrients available from the substrate were released in the remediated AMD water (Day 73) and provided an indication of the selected substrate's ability to support SRB during the $SO_4$ reduction process in the AMD.

**Table 9.** Nutrients measured in the water column after 72 days' exposure to AMD and different selected organic materials or waste materials.

| Substrate | % TOC | % N | % P |
|---|---|---|---|
| Cow manure (Control) | 0.13 | 0.18 | 1.71 |
| Chicken feathers | 0.27 | 0.13 | 0.02 |
| Cow manure and chicken feathers | 0.62 | 0.61 | 1.35 |
| Slimes and chicken feathers | 0.35 | 0.65 | 0.06 |
| Fly ash and chicken feathers | 3.62 | 1.93 | 0.13 |
| Sludge and chicken feathers | 0.13 | 0.56 | Not detected |
| Slag and chicken feathers | 0.02 | 1.29 | Not detected |
| Discard and chicken feathers | 0.04 | 0.29 | Not detected |

*3.2. Phase 1 Versus Phase 2 Substrate Efficiency and Remediation Achievement*

When assessing if the chicken feathers' substrate mixtures in Phase 2 were more efficient in the treatment of AMD per volume, the pH and $SO_4$ changes observed in the remediated AMD water samples (Day 35 for Phase 1 and Day 73 for Phase 2) were compared. The water pH levels of the remediated AMD increased throughout both experimental phases (Figure 7). Overall, the Phase 2 substrate mixtures achieved higher pH levels in the remediated AMD water, compared to the Phase 1 substrates (apart from the discard substrate). The variations between the Phase 1 and 2 experimental set-ups (AMD volume, substrate composition, substrate volume and exposure timeframe) must be considered to determine the performance of each substrate (Phase 1), compared to the chicken feathers mixed substrate (Phase 2).

The $SO_4$ reduction in the remediated AMD of Phase 2 improved for the sludge and chicken feathers, and slag and chicken feathers mixed substrates, with significant *p*-values being indicated. However, comparing these $SO_4$ reduction improvements to the performance in both experimental phases of the control cow-manure sample, the sludge and chicken feathers and slag and chicken feathers substrates did not show any improvements in $SO_4$ removal performance during Phase 2. However, the slimes and chicken feathers, and discard and chicken feathers mixed substrates showed a large improvement in $SO_4$ removal rates. These single substrates in Phase 1 increased the $SO_4$ concentrations in the remediated AMD water samples (Day 35) but significantly reduced the $SO_4$ concentrations during

Phase 2, with comparisons to the control (cow manure) substrate showing a large variation. The fly ash and chicken feathers substrate mixture also showed improved efficiency regarding $SO_4$ removal from the remediated AMD water (Day 73). The latter substrate mixture was the only substrate mixture that achieved a $SO_4$ reduction in the remediated AMD water in both Phase 1 and 2 and showed increased efficiency in terms of $SO_4$ removal from the remediated AMD water in Phase 2 based on the volume treatment. Furthermore, the increased pH and alkalinity levels resulted in a reduced metal concentration in all remediated AMD water samples.

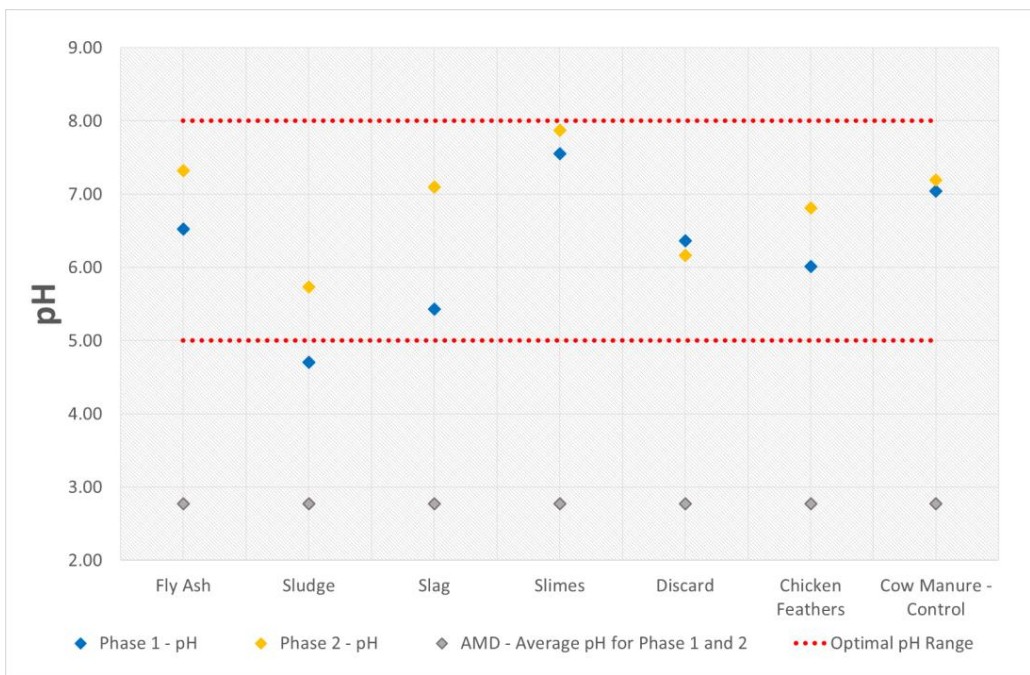

**Figure 7.** pH changes achieved in remediated AMD after exposing AMD water (Day 0) to substrates selected for Phase 1 (Day 35) and Phase 2 (Day 73).

Figure 8 shows the percentage of $SO_4$ reduction achieved within the remediated AMD for both Phases 1 and 2. When comparing the level of $SO_4$ reduction achieved by the various substrates of both Phase 1 and 2 static column experiments with the specific control (cow manure), the variations within the different phases (time exposure, substrate content and volume, AMD treated volume) were considered when assessing the various substrate's efficiencies in removing $SO_4$ from the remediated AMD. The variations between the experimental set-ups of Phases 1 and 2 resulted in a 24% higher $SO_4$ reduction rate for the control cow-manure sample in Phase 2 (42% in Phase 1 versus 66% in Phase 2), and a 30% higher $SO_4$ reduction rate for the chicken feathers sample in Phase 2 (10% in Phase 1 versus 40% in Phase 2) per volume of substrate.

### 3.3. Microbial Diversity—16S rRNA Sequencing

The 16S rRNA sequencing method adds value to the detection of microbial diversity within samples compared to the traditional culture methods which, by nature of the method, reduce the ability to detect microbial diversity [26]. The percentage distribution of the most abundant 16S rRNA OTUs sequenced (exceeding 1% presence) and the total number of taxa detected from the untreated AMD water (Day 0 AMD) and the remediated AMD water after exposure to the various substrates during Phase 2 (Day 73 AMD) are shown in Figures 9 and 10. The cumulative percentage of the number of taxa sequenced but detected with a <1% presence (not dominant) of the microbial diversity, are represented as a cumulative group, known as "Other spp." (other species). This comprised a significant number of the taxa detected within each sample, ranging from 12.5% in the sludge- and chicken-feathers-

remediated AMD (Day 73) to 45.1% in the cow-manure- and chicken-feathers-remediated AMD (Day 73). The dominant species (representing >1% in the various samples) ranged from 14 (sludge and chicken feathers; slimes and chicken feathers), 15 (slag and chicken feathers; fly ash and chicken feathers), 19 (cow manure and chicken feathers; cow manure alone; chicken feathers alone) and 24 (slimes and chicken feathers) OTUs detected in the remediated AMD samples (Day 73). The AMD as collected from the EWRP (Day 0) had 25 dominant OTUs present. The various taxa can be grouped into different microbial families and classes. This enables a better overview of the different microbial activities that may be responsible for bioremediation of AMD within the various static columns (Figure 10).

The only mixed remediated AMD sample (Day 73) that displayed more taxonomic classes than the cow manure control (seven classes) was the cow-manure- and chicken-feathers-remediated AMD (eight classes). From the classes detected through the 16S rRNA sequencing, the classes known for SRB activity are *Clostridia*, *Bacilli*, *Alphaproteobacteria* and *Delta proteobacteria* [32–35]. Microbial classes known for their ability to reduce nitrate include *Betaproteobacteria*, *Actinobacteria*, *Cytophagia*, *Gammaproteobacteria* and *Haloplasmataceae* [33,34]. The *Flavobacteriia*, *Haloplasataceae* and *Sphingobacteriia* classes are capable of utilising amino acids to support growth and sustainability [33,34]. Reviewing the various dominant microbial family distributions within the remediated AMD samples, the majority consisted of SRB, followed by nitrogen-reducing bacteria (NRB) (Figures 11 and 12). The two families that were present in all remediated AMD water samples, and the AMD untreated water sample were the *Clostridiaceae* and *Peptococcaceae* families. The substrates that supported the lowest microbial familial biodiversity were the sludge and chicken feathers mixture, and the slag and chicken feathers mixture (5 and 4 represented microbial families, respectively). The discard- and chicken-feathers-, slimes- and chicken-feathers-, fly-ash- and chicken-feathers-, and chicken-feathers-remediated AMD water (Day 73) revealed similar microbial diversity levels (8, 9, 8 and 9 represented microbial families, respectively).

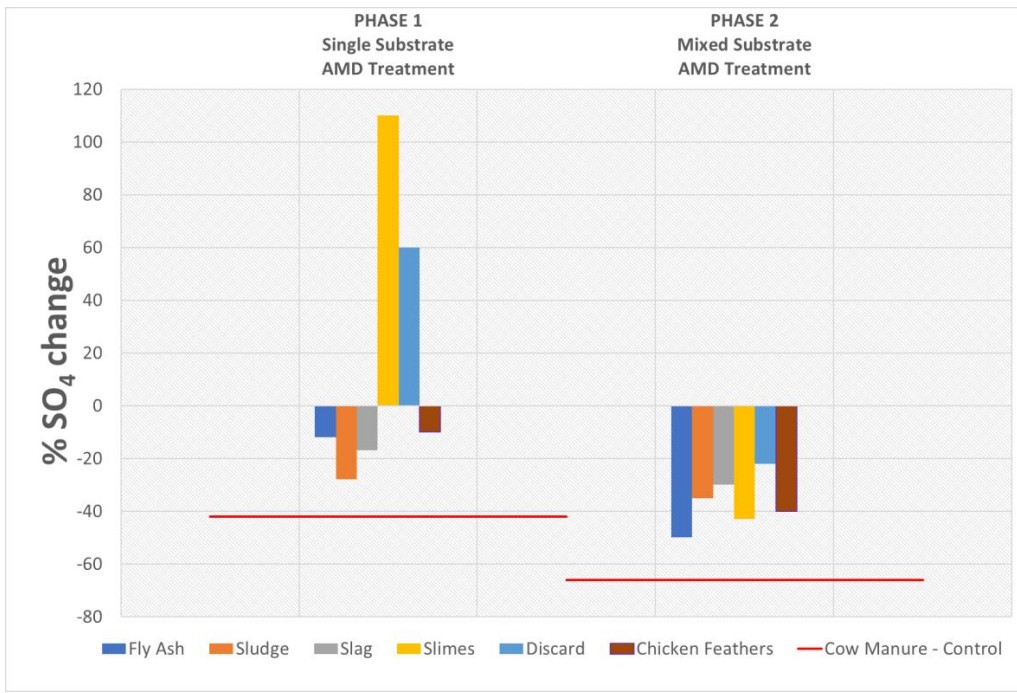

**Figure 8.** The average percentage change in AMD sulphate content achieved during Phase 1 (Day 35) and 2 (Day 73).

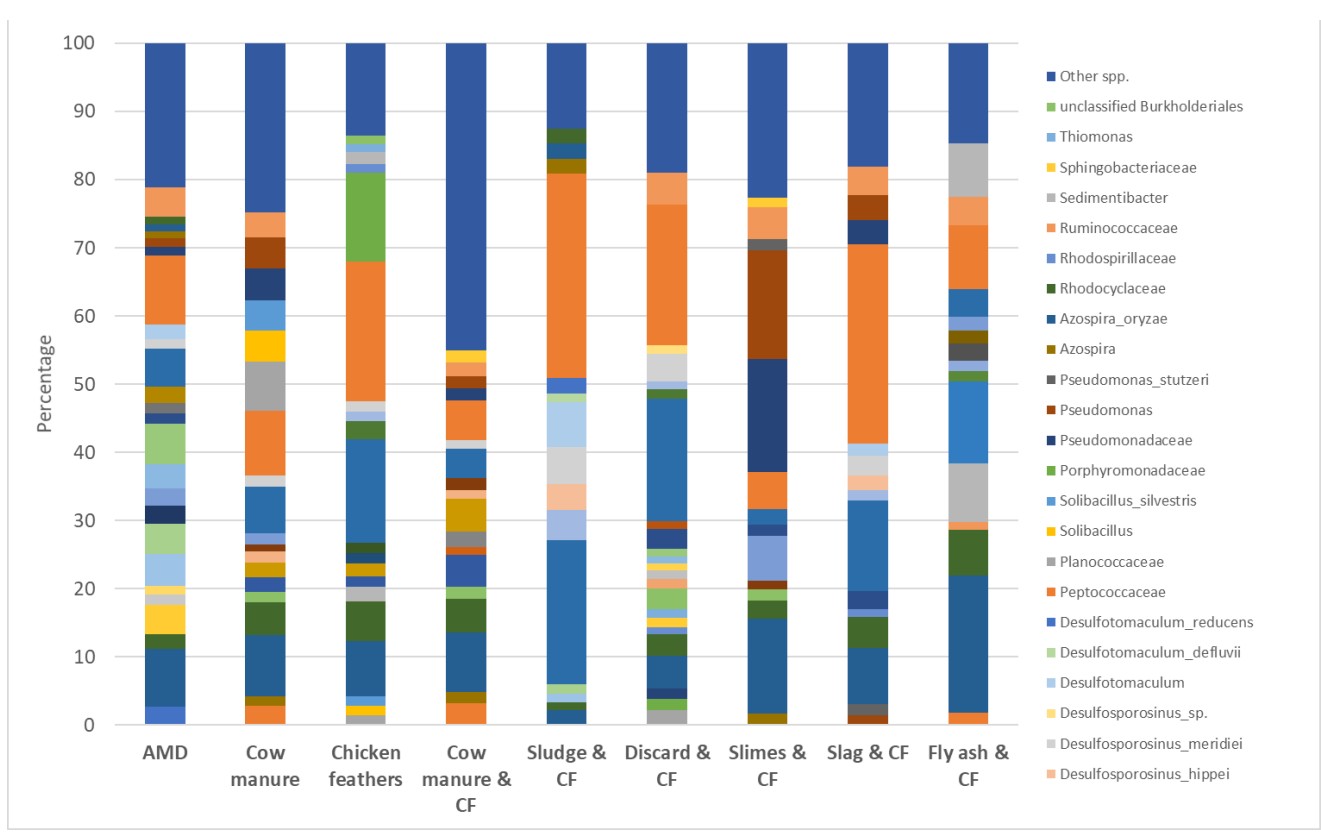

**Figure 9.** Percentage distribution of taxa detected within the selected AMD samples (Day 0 and Day 73 AMD water samples).

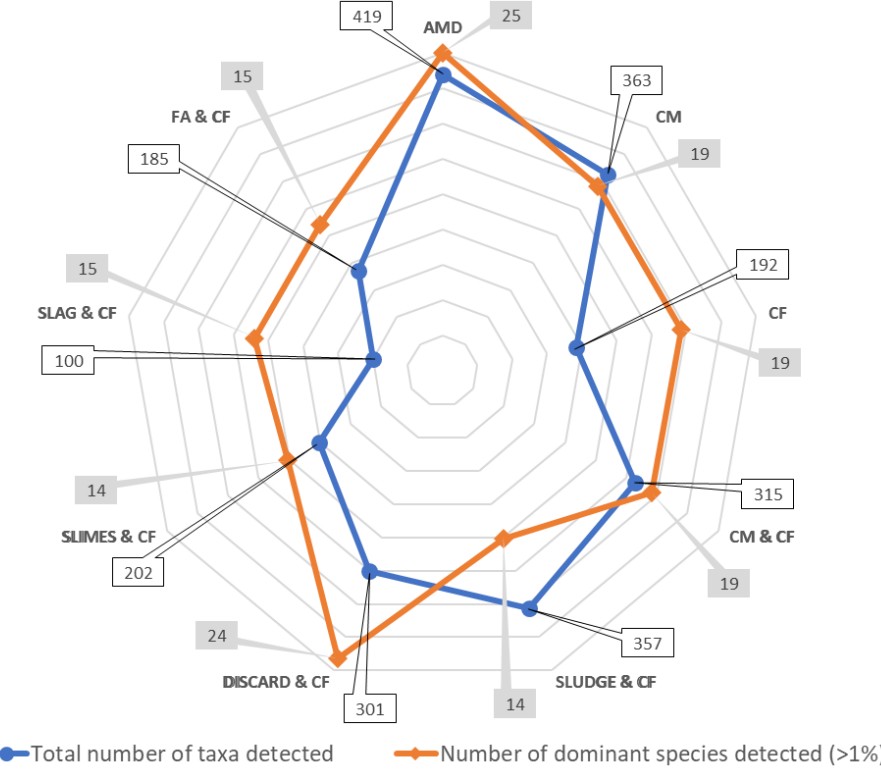

**Figure 10.** Total number of taxa detected versus the number of dominant species (>1% representation) per sample.

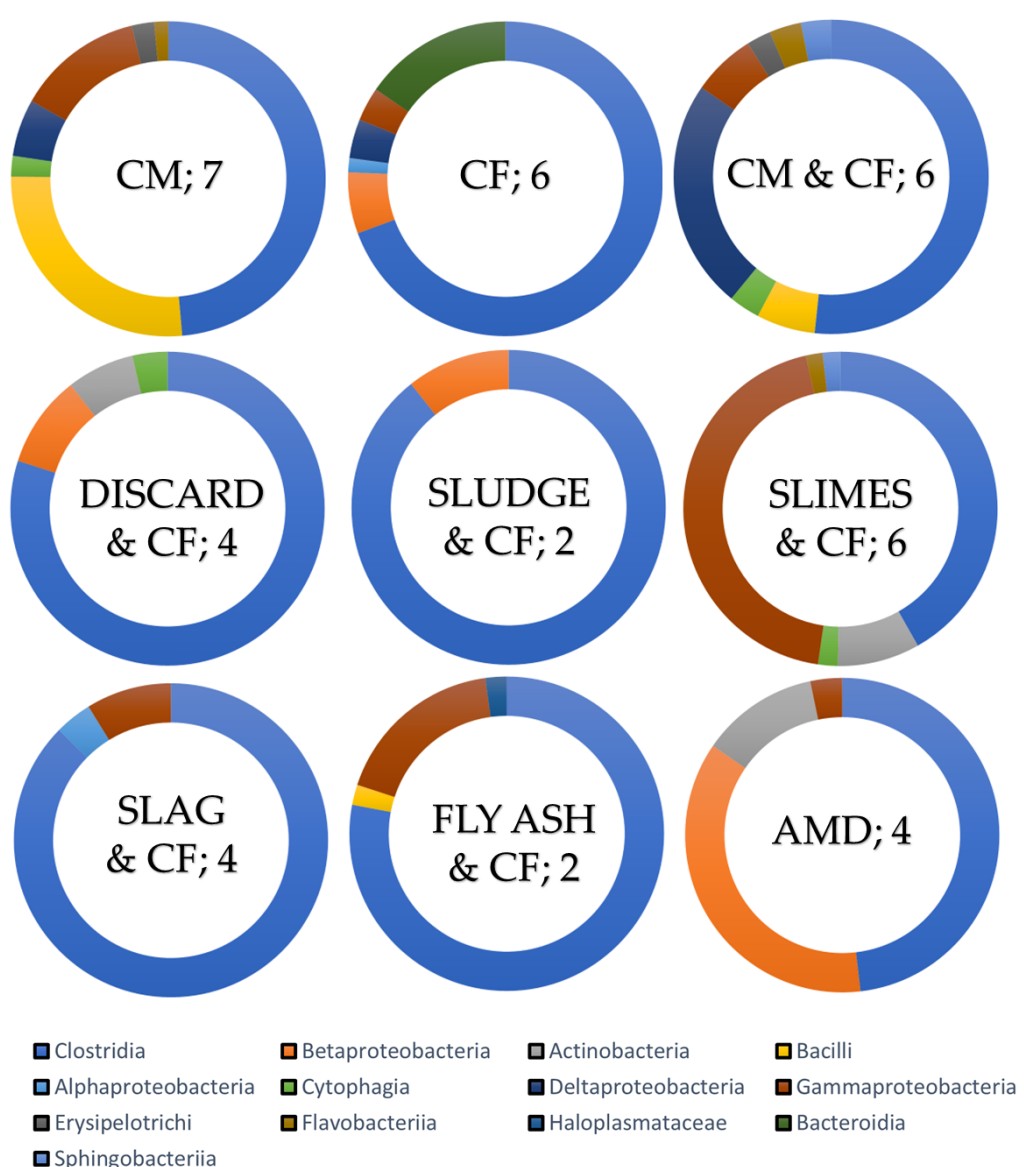

**Figure 11.** Microbial taxonomic classes sequenced with 16S rRNA technology within the remediated AMD water samples from Phase 2 (Day 73).

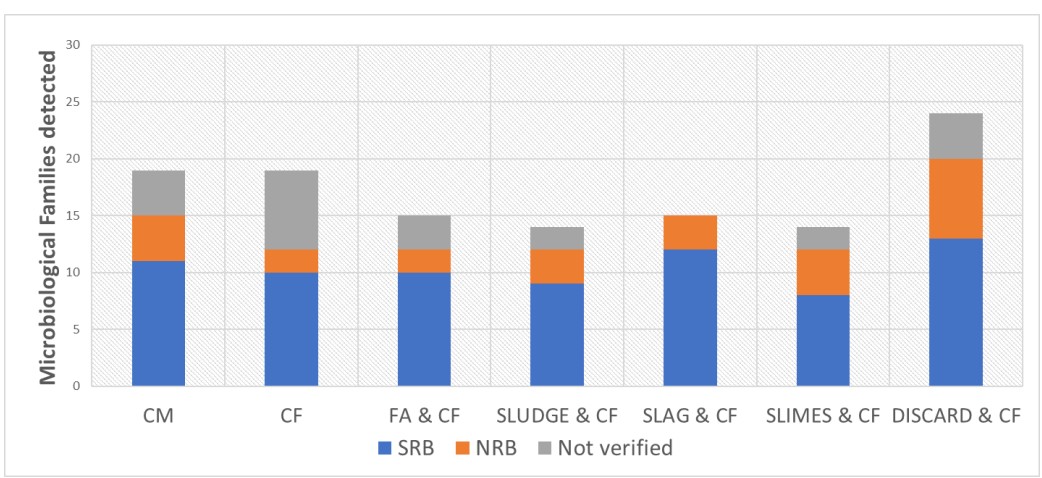

**Figure 12.** Presence of sulphate-reducing bacteria, nitrogen-reducing bacteria, and unverified microbial reduction activities.

### 3.4. Principal Component Analyses

From the PCA (Figure 13), a positive correlation was found between the cow-manure-containing substrates and the water parameters of orthophosphate, alkalinity and TDS. There was also a strong association between the *Erysipelotrichi* and *Bacilli* microbial classes, of which the *Bacilli* class is known to be associated with $SO_4$ reducers. The *Deltaproteobacteria* class also contained SRB and occurred in association with the *Gammaproteobacteria* class, which was NRB [33,34]. These classes are closely associated with elevated pH levels, $NH_3$ and $NH_4$ and manganese concentrations in the water chemistry of the remediated AMD water of Phase 2 (Day 73). The slag and chicken feathers, fly ash and chicken feathers, and chicken feathers substrate-remediated AMD water are associated with *Haloplasmataceae*, which are also known for their N-reducing ability. *Alphaproteobacteria* (a class known to have both SRB and NRB genuses) were also associated with these substrates (31–32). The sulphate reducers in the *Clostridia* class were also associated with these substrates, but to a lesser extent than the latter two microbial classes.

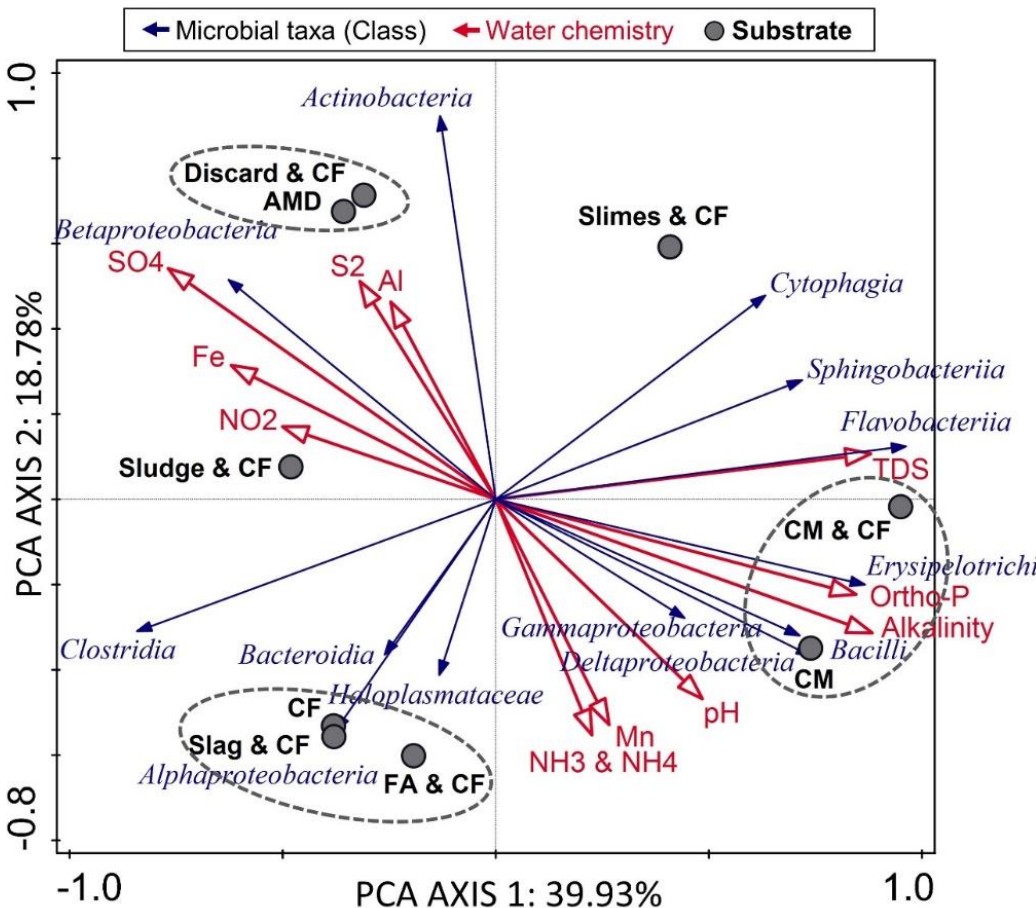

**Figure 13.** Principal component analyses triplot, indicating the associations between the selected substrate mixtures, water quality parameters and various microbial taxa detected.

The $NH_3$ and $NH_4$ and manganese water parameters of the remediated AMD water in Phase 2 (Day 73) have a similar association with the fly ash and chicken feathers, as with the cow manure substrate. *Actinobacteria* and *Betaproteobacteria* are NRB and are associated with increased $SO_4$ and Al levels in the water chemistry of the remediated AMD water in Phase 2 (Day 73), as well as the discard and chicken feathers substrate mixture. The microbial diversity that was most closely associated with the slimes and chicken feathers mixture included the *Cytophagia* class (NRBs).

Skousen et al. [19] identified that microbial diversity in passive treatment systems is important as it aids in the establishment of microbial communities. These communities

work together to produce simple organic compounds from the substrates that are present, which, in turn, can support SRB, aiding in the passive treatment of the AMD. Such systems are complex and must be monitored to understand the impacts they may have on the passive treatment system. De Klerk et al. [18] discussed other important factors in passive treatment systems such as wetlands, which include flow rate, sedimentation and filtration ability, with each affecting the microbial communities in these systems. The species' richness, expressed as OTUs, detected in the AMD water (Day 0) was the highest, with 25 different taxa represented (Figure 10). A definite microbial community shift in the remediated AMD water (Day 73) was observed when it was exposed to the various substrate mixtures selected for Phase 2. The PCA provided insight into the microbial community shift that occurred in the remediated AMD water (Day 73), highlighting the associations between the typical microbial classes supported by the various selected substrate mixtures. Microbial classes linked to SRB presence were found to be associated with the cow-manure-containing substrates, the chicken feathers substrate and two of the chicken feathers mixed substrates (the slag and fly ash substrate mixtures).

## 4. Conclusions

The selected substrates for both Phase 1 and 2 static experimental columns achieved some form of AMD water treatment, either through pH elevation, $SO_4$ reduction or metal removal from the remediated AMD water. Mixing the various substrates with recycled chicken feathers elevated the carbon content of the mixed substrates, potentially increasing the substrates' ability to support microbial activity. An association with SRB was shown in the *Clostridia* and *Alphaproteobacteria* classes and the chicken feathers substrate, which, over time, may support the SRB activities in passive treatment. Each substrate's performance was measured based on the volume used in treatment. Fly ash's ability to remediate AMD was improved in the fly ash and chicken feathers mixture during the Phase 2 experimental set-up. This was confirmed through a comparison of the $SO_4$ percentage change from the AMD to remediated AMD samples, relative to the cow manure control in both phases for the fly ash substrate. Similar achievements were noted for the slimes and chicken feathers, and discard and chicken feathers substrates.

Overall, the selected substrates and substrate mixtures with chicken feathers showed potential in initiating passive treatment for AMD, as increased pH and alkalinity, and reduced sulphate and metal contents, were observed in the remediated AMD water samples. It was evident from the study that the addition of chicken feathers to substrates aided in the enhancement of AMD water remediation through the significant improvements observed in pH elevation and sulphate reduction rates in the remediated AMD samples. Microbial diversity was also supported by the various substrates that included SRB, although this presence did not indicate an association with any remediated AMD water parameters in the slag and chicken feathers, fly ash and chicken feathers, and chicken feathers substrates. The extended timeframe of 73 days at which these mixed substrates were exposed to AMD showed that the various chicken feathers mixed substrates can support microbial activity and maintain passive remediation activities in AMD water. These substrates' availability to SRB to support the anaerobic activities associated with AMD remediation over longer timeframes, however, has yet to be determined. The assessment was primarily conducted based on volumes and was not completed relative to the mass of the different substrates, although these were accounted for. Considering the different densities of the selected substrates, the carbon content relative to the mass of the substrate and its ability to initiate passive treatment on the AMD should be confirmed.

**Author Contributions:** Conceptualization, A.-M.B. and P.J.O.; Data curation, J.C.T. and A.-M.B.; Formal analysis, M.M.O., J.C.T. and A.-M.B.; Funding acquisition, P.J.O.; Investigation, M.M.O. and L.L.N.; Methodology, M.M.O. and L.L.N.; Project administration, P.J.O.; Software, J.C.T. and P.J.O.; Supervision, L.L.N., A.-M.B. and P.J.O.; Writing—original draft, M.M.O., A.-M.B. and J.C.T.; Writing—review and editing, M.M.O., L.L.N., J.C.T. and P.J.O. All authors have read and agreed to the published version of the manuscript.

**Funding:** The research was funded by Coaltech Research Association: Grant No. E2020/6.

**Institutional Review Board Statement:** Not applicable.

**Informed Consent Statement:** Not applicable.

**Data Availability Statement:** All relevant data presented in the article are stored according to institutional requirements and, as such, are not available online. However, all data used in this manuscript can be made available upon request to the authors.

**Conflicts of Interest:** The authors declare no conflict of interest.

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
