# Peer review of "Assessing Alternative Supporting Organic Materials for the Enhancement of Water Reuse in Subsurface Constructed Wetlands Receiving Acid Mine Drainage"

_recycling, doi:10.3390/recycling7030041_

Round 1

Reviewer 1 Report

The manuscript " Assessing alternative supporting organic materials for the enhancement of water reuse in subsurface constructed wetlands receiving acid mine drainage" evaluted the enhanced effects of different organic waste on AMD treatment by constructed wetland. The author provide sufficient data to proved that chicken feather (powder) could amened the ADM tratment by increasing the pH, promoting the sulfate reduction and support the growth of microbial. This results is interesing and convincing, which might enable the use of chicken feather in selected bioprocesses. It is fine the publish this article in Recycling after minor revison.

  1. Figure 1b: Please check the unit of EC
  2. Figure 4: The legend is not provided

Author Response

Reviewer 1 comments

Corrections made

1

Check EC unit in figure 1b

Figure 1b amended to display the correct unit for EC as mS/cm

2

Provide legend for figure 4

It is assumed that the referenced comment is directed towards the image tabled as part of table 1.  Table image and the image was presented as the new  was therefore amended to remove the figure as part of the table.  The image was then listed as Figure 4 which resulted in the subsequent figure captions being renumbered.  References in text to the various amended figures were also added and amended to refer to the correct figure number due to this change.   

Reviewer 2 Report

The proposed article can be interesting for specialists in the area of acid mine drainage and groundwater treatment. The subject suits the scope of Recycling. There are some specific comments:

1.       Introduction recommended to update with information about the treatment of groundwater with high organic matter content with passive techniques (permeable reactive barriers). Following publications can be used for this:

https://doi.org/10.1016/j.jece.2017.06.011

https://doi.org/10.1016/j.jece.2015.04.017

In general, the manuscript is interesting and well written. It can be accepted with minor revision.

Author Response

Reviewer 2 comments

Corrections made

1

Refine the introduction to include information on treatment high organic matter in groundwater using passive techniques

Added sentence to introduction listing passive permeable reactive barriers as a passive treatment option for contaminated groundwater.  Added references as numbers 34 and 35 on the reference list.